# Consensus Recommendations for the Clinical Management of Hematological Malignancies in Patients with DNA Double Stranded Break Disorders

**DOI:** 10.3390/cancers14082000

**Published:** 2022-04-14

**Authors:** Agata Pastorczak, Andishe Attarbaschi, Simon Bomken, Arndt Borkhardt, Jutte van der Werff ten Bosch, Sarah Elitzur, Andrew R. Gennery, Eva Hlavackova, Arpád Kerekes, Zdenka Křenová, Wojciech Mlynarski, Tomasz Szczepanski, Tessa Wassenberg, Jan Loeffen

**Affiliations:** 1Department of Pediatrics, Oncology and Hematology, Medical University of Lodz, 91-738 Lodz, Poland; wojciech.mlynarski@umed.lodz.pl; 2Department of Pediatrics, Pediatric Hematology and Oncology, St. Anna Children’s Hospital, Medical University of Vienna, 1090 Vienna, Austria; andishe.attarbaschi@stanna.at; 3Department of Pediatrics and Adolescent Medicine, Medical University of Vienna, 1090 Vienna, Austria; 4Great North Children’s Hospital, Newcastle upon Tyne Hospitals NHS Foundation Trust, Newcastle upon Tyne NE7 7DN, UK; s.n.bomken@newcastle.ac.uk (S.B.); andrew.gennery@newcastle.ac.uk (A.R.G.); 5Translational and Clinical Research Institute, Newcastle University, Newcastle upon Tyne NE2 4HH, UK; 6Department of Pediatric Oncology, Hematology and Clinical Immunology, University Children’s Hospital, Medical Faculty, Heinrich Heine University, 40225 Düsseldorf, Germany; arndt.borkhardt@med.uni-duesseldorf.de; 7Department of Pediatric Hematology, Oncology and Immunology, University Hospital Brussels, 1090 Jette Brussels, Belgium; jvdwerff@uzbrussel.be; 8Pediatric Hematology-Oncology, Schneider Children’s Medical Center, Petach Tikvah 4920235, Israel; sarhae@clalit.org.il; 9Department of Pediatric Oncology, University Hospital and Faculty of Medicine, Masaryk University, 662 63 Brno, Czech Republic; e.hlavackova@fnusa.cz (E.H.); krenova.zdenka@fnbrno.cz (Z.K.); 10Department of Clinical Immunology and Allergology, St. Anne’s University Hospital in Brno, Faculty of Medicine, Masaryk University, 662 63 Brno, Czech Republic; kerekes.arpad@fnbrno.cz; 11Department of Pediatric Hematology and Oncology, Medical University of Silesia (SUM), 41-800 Zabrze, Poland; szczep57@poczta.onet.pl; 12Department of Neurology and Child Neurology, Donders Institute for Brain, Cognition and Behavior, Radboud University Medical Center, 6525 GA Nijmegen, The Netherlands; tessa.wassenberg@uzbrussel.be; 13Princess Máxima Center for Pediatric Oncology, 3584 CS Utrecht, The Netherlands; j.l.c.loeffen@prinsesmaximacentrum.nl

**Keywords:** DNA repair disorder, Ataxia Telangiectasia, Nijmegen breakage syndrome, leukemia, lymphoma, clinical management

## Abstract

**Simple Summary:**

Ataxia Telangiectasia (AT) and Nijmegen breakage syndrome (NBS) are the most common DNA repair disorders (DNARDs), characterized by an exceedingly high risk for developing hematological malignancies and poor outcomes. Clinical management of lymphoproliferative diseases in AT and NBS is complicated due to the competing challenges of delivery of optimal cancer treatment and management of excessive toxicities. AT and NBS are rare genetic entities in the general population, thus gaining extensive experience in treatment of these patients is difficult. Additionally, no treatment guidelines for lymphoproliferative diseases have been specifically designed for this group of patients as yet. In this review we formulate clinical recommendations, considering the most critical aspects related to the management of lymphoproliferative disorders in AT and NBS and we concisely present the current state of knowledge about the biology and outcomes of leukemia and lymphoma in these DNARDs.

**Abstract:**

Patients with double stranded DNA repair disorders (DNARDs) (Ataxia Telangiectasia (AT) and Nijmegen Breakage syndrome (NBS)) are at a very high risk for developing hematological malignancies in the first two decades of life. The most common neoplasms are T-cell lymphoblastic malignancies (T-cell ALL and T-cell LBL) and diffuse large B cell lymphoma (DLBCL). Treatment of these patients is challenging due to severe complications of the repair disorder itself (e.g., congenital defects, progressive movement disorders, immunological disturbances and progressive lung disease) and excessive toxicity resulting from chemotherapeutic treatment. Frequent complications during treatment for malignancies are deterioration of pre-existing lung disease, neurological complications, severe mucositis, life threating infections and feeding difficulties leading to significant malnutrition. These complications make modifications to commonly used treatment protocols necessary in almost all patients. Considering the rarity of DNARDs it is difficult for individual physicians to obtain sufficient experience in treating these vulnerable patients. Therefore, a team of experts assembled all available knowledge and translated this information into best available evidence-based treatment recommendations.

## 1. Introduction

Ataxia Telangiectasia (AT) and Nijmegen breakage syndrome (NBS) are the most common DNA repair disorders (DNARDs) characterized by a very high risk for developing leukemia and lymphoma during childhood [1,2]. No general multidisciplinary treatment guidelines for lymphoproliferative diseases have been specifically designed for these vulnerable patients. This is mainly due to the variability of intra- and interindividual chemotherapeutic toxicity in this patient group. One patient can tolerate a specific course very well, and subsequently develops unexpected life-threatening complications after a similar course later in their treatment. These factors make the design of common treatment guidelines challenging.

In the case of hematological malignancies, patients with AT and NBS need a modified and individualized approach to the diagnostic process, appropriate therapy, treatment of co-morbidities and surveillance. The competing challenges of excessive toxicities and frequent cancer progression result in a poor outcome for this group of patients [3,4]. Since AT and NBS are rare genetic entities in the general pediatric population, gaining extensive experience in clinical management of these patients is very difficult. We believe that consensus clinical guidelines addressing diagnosis and treatment of lymphoproliferative diseases in these DNARDs may provide valuable knowledge for pediatric oncologists. Therefore, we brought together experts working on different aspects associated with leukemia and lymphoma in AT and NBS, who are collaborating within the LEGEND COST Action and Host Genomic Variation I-BFM Study Group. The group contains immunologists, pediatric hematologists, transplant physicians, neurologists and pulmonologists. Our first aim was to concisely present the current state of knowledge about the biology of leukemia and lymphoma in DNARDs and secondly to address the most critical aspects related to the clinical management of lymphoproliferative disorders in them.

## 2. Article Design

This review provides a practical guideline for the multidisciplinary treatment of patients with DNARDs who develop hematological malignancies. Each section defines the topic that will be addressed, followed by evidence-based or expert-based recommendations (highlighted in bold). If the level of evidence is lower than level VII (expert opinion which based on the best available literature) this is listed below the recommendation. The topics discussed are clinical phenotype; genotype; pathophysiology of leukemia and lymphoma; immunology; pulmonology; nutrition; anesthetic and perioperative risk; neurology; treatment of hematological malignancies; immunotherapy; the role of hematological stem cell transplantation and screening.

## 3. Clinical Phenotype of AT and NBS

The most prominent clinical features of classic AT are signs caused by progressive, severe neurodegenerative disease. Patients typically present with cerebellar ataxia, generally before the age of four years. Extrapyramidal movement disorders develop during childhood, while polyneuropathy and motor neuron disease occur in the second decade of life [5]. This combination leads to progressive motor dysfunction with patients often becoming wheelchair bound before adolescence. Additional neurological signs are oculomotor apraxia, dysarthria, postural changes and joint contractures. Although intelligence is typically normal, developmental delay, resulting partly from slow motor and verbal responses, occur frequently. Other important clinical features of AT are growth retardation, ocular cutaneous telangiectasias, progressive lung disease, premature ageing, diabetes mellitus and premature ovarian failure. Characteristic laboratory findings are elevated alfa-fetoprotein (AFP) levels, increased chromosome instability and radiosensitivity [6].

In contrast, patients with NBS are frequently affected by progressive and disproportionate microcephaly, dysmorphic facial appearance, growth retardation and moderate intellectual disability [1]. Microcephaly is already present in newborns with NBS and may progress to become severe in some patients [7]. However, microcephaly may be hidden by development of hydrocephalus or large subarachnoid cysts [8]. Craniofacial features become more pronounced with age and include a sloping forehead, upward slanted palpebral fissures, prominent nose, relatively large ears and a receding mandible [1]. Growth retardation is marked during the first three years of life, but it steadily improves. Developmental milestones are frequently met during the first year, but subsequent speech delay is common [7,9]. Intelligence may be normal during early childhood, but a gradual worsening of IQ usually occurs during childhood, resulting in mild to moderate intellectual disability [9]. Patients may also be affected by skin abnormalities such as hyperpigmented or hypopigmented macules and sarcoid-like granulomas [10]. Other congenital malformations are rare but include anomalies of the central nervous system (e.g., hydrocephaly, schizencephaly, arachnoid cysts), choanal atresia, cleft lip and palate, tracheal hypoplasia, polydactyly, horseshoe kidney, hydronephrosis, hypospadias, anal stenosis/atresia and congenital hip dysplasia [11].

Whilst clear differences exist in clinical phenotype and progression, AT and NBS also share central clinical features, notably (progressive) immunodeficiency, strong radiosensitivity and a significantly increased risk of malignancy. Immunodeficiency shows marked variability, both in terms of immune compartments affected as well as severity [12,13]. Overall, patients with NBS and AT are predisposed to recurrent pneumonia, bronchitis, sinusitis, otitis media and mastoiditis.

Both AT and NBS carry a significantly increased susceptibility to develop malignancies, especially of hematological origin [1,14,15] with lifetime accumulated risk estimated to be 10–30% and up to 70%, respectively. Acute lymphoblastic leukemia (ALL), Hodgkin lymphoma (HL) and non-Hodgkin lymphomas (NHL) account for approximately 90% of malignancies in DNARDs patients [4,16]. Equally, AT and NBS patients may present with non-malignant, clonal T-cell proliferations which harbor chromosomal rearrangements, mainly between chromosomes 7 and 14. Due to TCR dysregulation these clones may evolve to aggressive T-cell prolymphocytic leukemia (T-PLL) [17]. Other relatively commonly described tumors in AT patients are carcinomas (breast, gastric, thyroid, liver) and gliomas [4], whilst in NBS, rhabdomyosarcoma, thyroid carcinoma, gonadoblastoma, glioma, meningioma, neuroblastoma and Ewing sarcoma have been reported [1]. The clinical hallmarks of patients with AT and NBS are listed in Figure 1.

## 4. Genotype of AT and NBS

**NBS** All patients clinically diagnosed with NBS harbor biallelic (homozygous or rarely compound heterozygous) mutations in the *NBN* gene encoding the nibrin protein. Nibrin acts by localizing MRN complex members to sites of DNA double-strand breaks (DSB). It activates cell cycle checkpoints during DNA recombination and meiosis and maintains telomere integrity [18,19,20,21].

All described pathogenic variants are located within exons 6–10 of *NBN*. The most common 5-bp deletion (c.657_661del5, p.Lys219Asnfs) affects approximately 90% of all NBS patients. It represents the founder mutation in those of Slavic ancestry and affects approximately 70% of patients of North American ancestry [11,22]. This deletion and most other variants lead to the synthesis of truncated (partially functional) proteins by introducing either a premature stop codon or an alternative initiation of internal translational start site [23]. The most common loss-of-function deletion, c.657_661del5, and other hypomorphic truncating variants, are associated with the classical NBS presentation, while the rare missense mutation c.643C>T (p.Arg215Trp) contributes to the severe phenotype of NBS seen in compound heterozygous patients (c.657_661del5/c.643C>T) [8]. In contrast, homozygous carriers of c.741_742dupGG (p.Glu248GlyfsTer5), c.330T>G (p.Tyr110Ter) and c.1125G>A (p.Trp375Ter) truncating variants show milder features of NBS [24,25].

**AT** The syndrome is caused by the presence of biallelic (homozygous or compound heterozygous) pathogenic variants in the *ATM* gene encoding a large phosphoinositidyl-3-kinase-related protein kinase (PIKK). ATM is a central regulator of the cellular response to DNA damage and to telomere dysfunction [26,27,28].

More than 800 different mutations scattered throughout the *ATM* gene have been reported [29]. Recurrent *ATM* lesions associated with the founder effect are described in several countries and populations [30,31,32]. There are two main categories of *ATM* mutations: ‘’classic’’ and “atypical” (“variant” or “mild”) [2,33]. The first type includes truncating mutations leading to the loss of ATM expression and loss of ATM kinase activity. These defects are associated with the severe (classic) phenotype of AT. Atypical lesions include missense mutations, in-frame or leaky splice-site mutations which generate residual amounts of functioning ATM protein with a certain level of kinase activity. In these cases, clinical presentation of AT is usually mild, and the diagnosis of cancer can precede the diagnosis of AT [2]. The current knowledge about the correlation between *ATM* genotype and phenotype was recently summarized by Putti S et al. [34].

The allelic heterogeneity in *ATM* contributes to the different levels of ATM activity, and results in a pronounced clinical heterogeneity of AT patients [35]. Pediatric lymphoid malignancies in AT patients are almost exclusively associated with the loss of ATM kinase activity and the expression of residual ATM kinase activity has a strong protective effect against tumor development in children [34]. The correlation between AT genotypes and cancer risk has been established for a limited number of *ATM* variants, including those linked with a strong predisposition to malignancy (c.1A>G, c.7271T>G, c.8147T>C, c.8494C>T, c.6679C>T) and those associated with a relatively low risk (c.5762-1050A>G) [36,37,38,39,40].

## 5. Pathophysiology of Leukemia and Lymphoma in AT and NBS

Alterations in DNA damage response pathways are hallmarks of cancer. Both ATM and NBN proteins play crucial roles in the maintenance of genomic integrity. Biallelic loss of ATM and NBN functions result in an increased level of spontaneous chromosome breakages in lymphoid precursors and peripheral lymphocytes in patients with AT or NBS [41]. It has been known for many years that AT and NBS are associated with substantially higher incidence of malignancy compared with the normal population. Amongst early cohort-based studies it was apparent that lymphoid malignancy predominated and disproportionately affected children [15,42,43]. More recently, European-wide retrospective studies have identified both AT and NBS as prevalent predisposition syndromes amongst children with either ALL or NHL [44,45]. As patient registries have developed, reducing the impact of case-ascertainment bias, our understanding of the cancer spectrum and incidence has improved. Amongst 296 consecutive genetically confirmed cases of AT in the UK and The Netherlands, 66 people developed 47 lymphoid (36/47 aged ≤16 years of age) and 19 non-lymphoid tumors [35]. Analysis of the French CEREDIH registry identified 69 patients developing cancer within a cohort of 279 people with AT, resulting in a cumulative incidence of malignancy of 22.6% at 20 years and 38.2% at 40 years [4]. Sixty-one lymphoid malignancies were identified, with a further eight carcinomas and one glioma. The analysis of the European Society for Immunodeficiencies registry identified 80 cancers developing in 63/149 (42%) of registered cases of NBS [13]. A recently published study among two-hundred and forty-one (*n* = 241) patients with NBS enrolled from eleven countries, showed that 151 (66.8%) patients developed cancer with a median age at presentation of 9.3 years. Cumulative cancer incidence was 40.2 ± 3.5% and 77.8 ± 3.4% at 10 years and 20 years of follow-up, respectively. Regarding tumor type 62.9% patients were diagnosed with NHL, 21.2% patients developed acute leukemias predominantly of T-cell origin, 6.0% patients with HL and 7.2% patients were diagnosed with other solid malignancies [3].

Despite the many challenges which exist in making a diagnosis of lymphoma in this patient group [46], compounded by the historic lack of a dedicated histopathological classification of lymphoproliferative disorders in this group of patients, multiple studies have shown a disease spectrum distinct from that seen in sporadic pediatric cases. Most notable are the predominance of diffuse large B cell lymphomas (DLBCL), the relative absence of Burkitt lymphoma and, particularly in NBS patients and the high incidence of T-cell lymphoblastic leukemia/lymphoma [4,42,44,47,48] including peripheral T-cell lymphomas (PTCL) [3]. A further contrast is the relatively low incidence of EBV associated DLBCL [4,46] as compared to primary immunodeficiency or solid organ transplant associated lymphoproliferation where only a minority of pediatric cases would be expected to be EBV negative [49]. These findings suggest diverse mechanisms of lymphomagenesis including primary immunodeficiency and viral oncogenesis, reduced immune onco-surveillance and direct lymphoid specific DNA repair defects, e.g., immunoglobulin/T cell receptor loci recombination associated events. Inefficient elimination of damaged cells in NBS patients and the fact that nibrin is involved in class switch recombination (CSR) and the alternative end joining (alt-EJ) DNA repair pathway contribute to the specific susceptibility of B cells to genomic instability [50]. Moreover, a disturbance of V(D)J recombination, characterized by impaired resolution of RAG-induced IgH breaks, may promote formation of complex translocations involving the IgH locus in NBS lymphomas [51].

Intriguingly, initial genomic/transcriptomic studies suggest that distinct oncogenic processes may exist and indeed differ substantially between patients with AT and those with NBS [52]. The presence of chromothripsis, probably associated with telomere shortening, was observed in T-ALL cells from AT patients but it has not been found in either T-ALL or in other tumors in NBS patients. Finally, in addition to diverse oncogenic mechanisms, further clinical variation may arise from the functional outcome of different *ATM/NBN* mutations [5,35,53]. However, studies have shown both the presence [35,53] and absence [4] of an association between residual protein function and risk of lymphoid malignancy. Maciejczyk et al. demonstrated that regular substitution of antioxidants and free radical scavengers results in prolongation of the survival time of ATM or NBS deficient animal models and cell lines [54]. Furthermore, the authors showed that regularly substituted ATM deficient mice developed cancer significantly later, suggesting that oxidative stress plays a role in lymphomagenesis in these animals [2]. It is likely that additional, yet undefined, factors impact on the clinical heterogeneity within and between these disorders.

## 6. Immunological Abnormalities in Patients with DNARDs

NBS and AT belong to the group of combined immunodeficiencies associated with syndromic features [55,56]. The immunodeficiency results from the impaired double-strand break repair processes that affect V(D)J recombination and immunoglobulin class-switching, leading to clonally restricted immunoglobulin and T cell receptor rearrangement [57]. This subsequently leads to reduced or absent serum IgA and IgG2. However, the severity of immunodeficiency varies between patients from virtually no clinical signs to life threatening infections [5,58,59,60,61]. Additionally, immunological parameters may deteriorate with age in NBS patients, whereas only 10% of people with AT show progression of immune defects during their lifetime [2,13,62].

In AT and NBS patients the clinical manifestation of immunodeficiency consists of recurrent sinopulmonary infections and chronic bronchitis which may progress to bronchiectasis and in turn be complicated by pneumonia [2,62,63]. Clinical manifestations of immunodeficiency in patients with AT may predate the neurological symptoms. An increased risk of autoimmune or chronic inflammatory disease development and non-infective granulomatous disease was observed in both syndromes [1,2]. In time, the occurrence of lower respiratory tract infections increases in frequency as well as severity in patients with AT [64,65]. An inadequate cough, poor mucocilliary clearance and chronic aspiration together with an abnormal muscle tone can all contribute to this deterioration [2,64]. Increased occurrence of intracellular pathogens and opportunistic infections are rare in AT and NBS patients [13]. Sporadic cases of tuberculosis in NBS patients [1,57] as well as herpes zoster reactivation, Pneumocystis jiroveci pneumonia, candida bacteraemia and invasive aspergilloses have been described [2,62]. Severe or persistent warts, suggesting poor handling of papillomaviruses have been described in AT patients as well [2,63]. Colonization of the lower respiratory tract with pathogenic gram-negative bacteria as well as higher incidence of viral skin infections is seen in association with the AT-HIGM (hyper-IgM) phenotype [2].

Patients with AT and NBS may express several abnormal laboratory immunological parameters and functional tests (Table 1).

For AT patients, diverse abnormalities in immunoglobulin levels or subclasses have been described. AT patients may also express an abnormal humoral response to vaccinations resulting in a compromised immune reaction following contact with the specific pathogens, particularly polysaccharide-encapsulated bacteria [1,2,13,61]. Of special importance is the HIGM phenotype, hallmarked by elevated or normal IgM levels together with decreased IgG [58]. This HIGM phenotype increases the risk of severe infections and early mortality [58].

IgM or IgG monoclonal gammopathies occur in NBS patients with a high frequency [13,56]. Increasingly clonal rearrangements of BCR and/or TCR genes in peripheral blood lymphocytes, frequently accompanied by increasing EBV viral load, can be observed in some NBS patients and can precede the development of a lymphoid malignancy [1,51]. The decreased level of total IgG in these patients, may mask the presence of such an IgG gammopathy [13]. Patients with AT or NBS who express low or absent IgG levels usually regularly receive human immunoglobulin replacement therapy, according to international guidelines [66].


**All patients with AT or NBS should have standard immunological screening at diagnosis of a hematological malignancy, at the end of treatment and subsequently annually. A standard immunological screening means level of IgG, IgM and IgA together with lymphocyte subpopulations (Level VI).**


In contrast to NBS, in AT patients the pattern of immunodeficiency observed in the first five years of life usually remains identical lifelong [67]. The AT study group recommends at least annual monitoring of total serum IgG, IgA, IgM and IgG subclasses, M-protein and lymphocyte phenotyping (T-cell, B-cell, and NK-cell analysis) [2]. Considering there is no published understanding about how the development and treatment of a hematological malignancy influences the immune system in patients with AT and NBS, we advise to establish a baseline profile at diagnosis and end of treatment of the malignancy, and subsequently an annual screening, according to guidelines [64]. Response to polysaccharide antigen should be evaluated in children with AT and NBS older than two years of age but this testing may not be reliable when performed at diagnosis of the malignancy [64]. Some experts argue the value of regular testing of monoclonal proteins, EBV DNA load, HBsAg, HCV RNA and TCR/BCR gene rearrangement in NBS patients, as screening investigations for hematological malignancy [1,13,57]. In our opinion, there is not enough evidence to recommend measurements of any of these parameters in routine clinical management.

**Antibiotic prophylaxis according to the local protocols should be applied in all patients with AT or NBS during treatment for a hematological malignancy. Additionally, prophylaxis against encapsulated pathogens (e.g., Hemophilus influenza, Streptococcus pneumoniae) should be administered to all patients who suffer from at least two, or one life-threatening, pulmonary bacterial infections during treatment** [13,64,67,68].

Azithromycin is preferred, especially in patients with bronchiectasis [59]. As an alternative first line of antibiotic therapy, amoxicillin, and trimethoprim-sulfamethoxazole may also be given [65]. Prophylaxis against viral, fungal and Pneumocystis jiroveci infections is indicated in severely immunodeficient patients. Currently, prophylaxis for Pneumocystis jiroveci (PJ) is recommended for most sporadic childhood leukemia and lymphoma patients during treatment. Antifungal prophylaxis is a topic of debate. Usually, antifungal therapy is recommended during treatment phases accompanied by severe or prolonged neutropenia. We recommend giving antifungal and PJ prophylaxis and performing routine antifungal testing according to local protocols for sporadic patients. Prophylaxis against herpes is indicated in cases of unusual severe or repetitive herpes infection.


**All AT or NBS patients with HIGM phenotype, or decreased IgG (serum IgG < 4 g/L or IgG < 2 SD below age adjusted means [64,65]) should receive immunoglobulin replacement therapy (IGRT) [66] (Level VI).**


IGRT is not indicated in IgG subclass deficiency with normal specific antibody responses [64]. Trough IgG serum levels should target at least 6 g/L in patients receiving IGRT [64] with potential end organ damage (such as bronchiectasis) [1,13,66].

## 7. Pulmonary Manifestation in AT and NBS

Progressive respiratory disease in patients with AT and NBS has a great impact on the patient’s quality of life. Optimal respiratory management should therefore be an integral part of supportive care from the time the diagnosis of AT and NBS is made. To date there are no specific recommendations for respiratory care of patients with DNARDs who develop hematological malignancies.

Encapsulated bacteria such as H. influenzae, Str. pneumoniae, and M. catarrhalis are the most common causative microorganisms of recurrent pulmonary infections in AT and NBS patients [69]. As lung damage progresses *P. aeruginosa* and *S. aureus* become more important pathogens [70]. The spectrum of non-infectious complications is broad: non-caseating granulomas, granulomatous-lymphocytic interstitial lung disease (GLILD), bronchiolitis obliterans organizing pneumonia (BOOP), follicular bronchiolitis (FB), lymphocytic interstitial pneumonitis (LIP) and lymphoid hyperplasia [71].

The presence of GLILD is associated with an increased incidence of malignancy [70]. This may cause a diagnostic challenge in distinguishing between non-malignant and malignant lymphoproliferative disease [72]. Standard of care treatment strategy of GLILD consists of induction by corticosteroids, which often does not result in durable remission. A combination of immunosuppressive therapy (rituximab, azathioprine, mycofenolate mofetil) may be then considered [69]. The lung serves as the reservoir of EBV. The EBV titre in sputum positively correlates with a decline in lung function in patients with bronchiectasis [73]. Certain B cell abnormalities affect the interaction of EBV with the host, may lead to chronic active EBV-infection (as in AT and NBS), and in this way increase the risk of bronchiectasis and loss of lung vital capacity.


**Thoracic MRI can be considered as first choice for diagnostics and follow up of acute or chronic respiratory disease in patients with DNARDs at diagnosis, treatment and follow-up of a hematological malignancy (Level VI).**


Patients with DNARDs have increased sensitivity to ionizing radiation [74]. This must be considered when planning imaging of patients with DNARDs who suffer from acute or chronic pulmonary disease [70]. In current medical practice, high resolution CT (HRCT) remains the gold standard to assess bronchiectasis as well as opportunistic pulmonary infections in immune compromised patients [75]. However, in recent years MRI has been widely considered to have replaced CT as a diagnostic procedure in several diseases including cystic fibrosis, primary ciliary dyskinesia and non-CF bronchiectasis [75,76,77].


**If age and the patient’s condition permits, pulmonary function tests should be performed prior to treatment of hematological malignancies, during therapy and at the end of treatment. In case of pulmonary deterioration during oncological treatment patients should promptly receive intensive physiotherapy and antibiotics.**


In general, for patients with DNARDs it is recommended to perform PFTs annually from the age of four onwards. Usually, there is a combination of an obstructive and restrictive pattern (latter sometimes more dominant). Multiple breath inert gas washout (MBW) testing with LCI (lung clearance index) correlates with CT severity score in non-CF bronchiectasis [77] and therefore provides an alternative measure of pulmonary function. Routine culturing of sputum is not usually performed in patients with DNARDs. This is mainly due to the difficulty of producing sputum. In addition, the specific influence of colonization of Pseudomonas species on reduced pulmonary function is a topic of debate. Most patients with bronchiectasis produce sputum on a daily basis [78]. In order to prevent progression of bronchiectasis and subsequent decreased pulmonary function, it is important to study the microbiology of recurrent infections. Protocols used by experts in care of chronic respiratory suppurating diseases may be applied. Bronchoscopy with bronchioalveolar lavage (BAL) may be helpful to identify structural anomalies and to obtain diagnostic sputum in children unable to expectorate and/or with unexpected decline of pulmonary function [77]. In symptomatic cases more frequent pulmonary function testing is indicated (3- or 6-monthly), but at the present time there is not a consensus screening protocol to monitor respiratory status in primary immunodeficiency [72].


**All patients with DNARDs who develop hematological malignancies should be counseled and guided by a respiratory therapist/physiotherapist.**


Effective airway clearance and antibiotic use are essential in chronic bronchitis and bronchiectasis management [78]. Therefore, the regular support of a physiotherapist who provides a personalized approach to selecting the most appropriate airway clearance technique can contribute to improved respiratory efficiency in patients with DNARDs. Additionally, effective airway clearance may be stimulated by hypertonic saline which alters the viscoelastic features of mucus, increases hydration of the airway–surface liquid and directly stimulates cough [79]. The use of hypertonic saline is well tolerated by most children: in case of bronchial hyper-reactivity, premedication with a bronchodilator may be needed [78]. Clinical studies assessing the effectiveness of N-acetyl cysteine administration have provided conflicting results [80]. The use of rhDNase is effective in CF but has been shown to be potentially harmful after prolonged use resulting in more frequent pulmonary exacerbations and greater FEV1 decline [79,81]. RhDNase should therefore be avoided in long-term management of non-CF bronchiectasis in pediatric patients [77]. Short-term use of rhDNase may improve respiratory failure with atelectasis [82]. Exacerbations of chronic bronchitis should be treated with antibiotics based on susceptibility testing [78]. Eradication of Pseudomonas aeruginosa often needs a combination of systemic and inhaled antibiotics. Anti-inflammatory and direct antimicrobial effects of azithromycin could be a new strategy in the prevention of exacerbations in the presence of bronchiectasis. For this indication, azithromycin should be given three times a week in a long-term setting [77].

## 8. Nutritional Status

Feeding problems are common in patients with DNARDs, especially after the first decade of life. Due to neurological deterioration, AT patients increasingly experience difficulties with chewing and swallowing, which may lead to decreased intake and subsequent malnourishment. These patients are more susceptible to aspiration. Aspiration is recognized by coughing and choking during eating, but can also be silent, and leads to recurrent pulmonary infections [5,83].


**A detailed growth curve should be reconstructed at diagnosis of a hematological malignancy in all patients with AT or NBS. Weight should be stringently followed on at least a weekly basis.**


The development of a hematological malignancy and the subsequent treatment puts these patients at risk of greater feeding difficulties which compound existing DNARDs induced malnourishment. The (unrecognized) cancer itself may present with weight loss. Immunochemotherapy may lead to nausea and alter taste which further decreases eating motivation. In addition, patients with DNARDs are prone to develop severe mucositis resulting in severe pain, malabsorption of nutrients, infectious complications and paresis of the gastrointestinal tract. These factors make patients with DNARDs who develop a hematological malignancy very susceptible to weight loss.


**All patients should be regularly counseled by a dietician. Additionally, patients with AT should be regularly counseled by a speech–language therapist with experience in treating patients with congenital language disorders.**


The focus should be on maintaining an optimal, stable weight during and following immunochemotherapeutic treatment. Stimulating oral intake and prevention of aspiration can be achieved by advising to take rest before meals, present food mashed or in small pieces in order to reduce excessive chewing exercise and the use of a straw [84].


**Since malnutrition and difficulty to thrive are frequent in patients with AT or NBS, nasogastric tube feeding and/or gastrostomy placement should be considered early in the cancer treatment process.**


A nasogastric tube, or in some specific cases gastrostomy placement (GT), is commonly necessary during treatment of sporadic hematological malignancies in order to maintain optimal nutritional status. Immunochemotherapeutic treatment exacerbates almost all DNARDs related complications and thereby negatively influences feeding and maintenance of optimal weight. Good tolerance of GT placement was reported in AT patients without cancer. They showed significant improvement in mealtime satisfaction and participation in daily activities [85]. However, the risk of GT-related complications including peristomal infections, bleeding or leakage may be increased in patients with DNARDs during intensive chemotherapy. In summary, the combination of DNARDs and hematological malignancy makes tube feeding or gastrostomy placement almost inevitable [83]. The optimal choice weighing risks and benefits should be made together with the patient and their caretakers.

## 9. Anesthetic and Perioperative Risk in Patients with AT and NBS

Patients with double stranded break disorders who develop hematological malignancies often have to undergo multiple invasive procedures e.g., placement of central venous line, bone marrow aspiration or biopsy, lumbar punctures and surgery due to tumor and treatment complications. These procedures are generally performed under general anesthesia or sedation.


**Patients with AT or NBS can usually receive general anesthetics safely. Preferably, this should be performed within specialist pediatric centers.**


Considering the progressive pulmonary dysfunction in patients with DNARDs, general anesthesia carries a risk which increases with age [86]. This is mainly due to the presence of GLILD (adolescent, young adult patients), the increased risk of aspiration and recurrent upper and lower respiratory tract infections. Progressive neurological impairment including swallowing difficulties, abnormal head posture and ineffective cough causes additional risk [87]. The precise role of these individual risk factors in patients with DNARDs needs to be further studied, especially in patients receiving oncological treatment. A retrospective study describing 21 patients with AT who underwent 34 anesthetics (median age 12.5 years) showed no major complications, no unplanned admissions and no mortality [86].

## 10. Neurological Aspects of Cancer Treatment in AT and NBS Patients

Leukemia and lymphoma can cause multiple neurological complications, either due to primary disease or because of treatment related toxicity [88,89]. In the case of AT, the diagnosis of the syndrome does not always precede the development of leukemia or lymphoma [90,91]. This is mainly due to fact that the neurological symptoms may become apparent and/or recognized in early childhood, while leukemia and lymphoma may be diagnosed during infancy. In contrast, patients with NBS are usually diagnosed at a younger age due to the associated dysmorphic features.


**Patients with AT or NBS often develop CNS abnormalities, thus, CNS MRI should be performed prior to the start of cancer treatment as a baseline.**


Cerebellar atrophy may be present in older children with AT on MRI scans of the brain. However, normal cerebral imaging is common in young children with AT and therefore it does not exclude the diagnosis of the syndrome [92,93].

Acute CNS complications during therapy of childhood leukemia and lymphoma arise in 10–15% of sporadic cases [94,95,96]. Long-term neuropsychological problems are often described as well [97]. Children with AT and NBS constitute a vulnerable group of patients with multisystem involvement and a progressive course of neurological impairment. It has not been systematically studied if children with AT or NBS are at increased risk for the specific neurological complications of leukemia and lymphoma.


**Patients with AT or NBS have a high incidence of CNS infection, therefore, patients treated for hematological malignancies should be investigated for CNS involvement early during an infectious episode.**


Due to their immunodeficiency, patients with AT and NBS have an increased risk for CNS infections during intensive chemotherapy [14]. Therefore, any evidence of neurological deterioration, especially during an episode of fever, should prompt early consideration of appropriate investigation for intracranial infection. This may include neuroimaging, diagnostic lumbar puncture or electro-encephalogram.


**Considering the progressive neurologic deterioration in patients with DNARDs, caution should be paid when using specific immunochemotherapeutic agents exhibiting a neurotoxic profile of side effects. In addition, in AT with pre-existing neurological symptoms, a 50% dose reduction for the neurotoxic agent vincristine or replacement by vinblastine should be considered.**


Patients with AT develop axonal sensorimotor neuropathy in the second decade of life. Although subtle electrophysiological abnormalities are already present earlier, clinical and electrophysiological polyneuropathy becomes evident around the time that patients become wheelchair bound [98]. It is not known if young children with AT have an increased risk for vincristine-related polyneuropathy. Uncomplicated use of vincristine in patients with AT has been described [99]. However, increased vulnerability seems likely and deterioration of neuropathy during vincristine use has been reported in multiple patients with AT [90,100,101,102]. An alternative approach is to substitute vinblastine for vincristine, which may cause less neurotoxicity [103,104].

There is insufficient evidence to advocate EMG surveillance during vincristine or vinblastine treatment [89], but at least regular neurological examination during chemotherapy with vincristine should be performed, similar to sporadic patients receiving vincristine.

Methotrexate can induce stroke-like episodes and leukoencephalopathy; however, it has been successfully used in treatment protocols in children with AT and NBS without excessive neurotoxicity [105,106].

Because of the importance of the neurological deterioration in patients with DNARDs and the possible acceleration of existing or new neurological problems during cancer treatment, a pediatric neurologist should be part of the multidisciplinary team treating the malignancy [64].

## 11. Treatment of Leukemia and Lymphoma in Children with AT and NBS

Each patient should have easy access to an oncologist. NBS patients should actively undergo follow-up in an oncological center. There should be a low threshold for blood investigations, MRI based imaging and subsequent surgical biopsy following presentation with clinical signs suspicious for cancer.

Treatment of leukemia and lymphoma in children with AT and NBS is extremely challenging. This is mainly due to the severe co-morbidities that accompany AT and NBS. Intensive anticancer treatment may (temporarily) worsen these co-morbidities whilst the severity of the co-morbidities may limit anticancer treatment intensity. Successful management requires a treatment schedule developed considering existing co-morbidities allied to a tailored package of supportive care from a broad multidisciplinary team. For example, a five-year-old girl with AT who develops T-ALL may have a good clinical condition with relative few co-morbidities, and treatment can be given with slightly modified treatment protocols; meanwhile, the same girl at sixteen years of age is likely to be wheelchair bound, to suffer from severely reduced pulmonary function and has progressive immunodeficiency. Treatment needs to be substantially modified in this patient with maximal supportive care instituted from the point of diagnosis. Treatment for similar malignancies can therefore be vastly different depending on a range of individual patient factors.

Genetic predisposition syndromes can influence the way the tumor responds to immunochemotherapy, as has been described for Li Fraumeni syndrome (hypodiploid acute lymphoblastic leukemia, with lower CR rates being reported) [107]. There is anecdotal evidence that this also applies for hematological malignancies in patients with DNARDs but isolating such an effect from the other challenges we describe here is complex. Carefully controlled studies need to be developed to investigate this further.

Considering the impact of cancer and anticancer treatment on the daily life of patients with AT and NBS, the wishes of patients and caretakers are extremely important in choosing the best individual treatment strategy. Only through true shared decision-making can we establish optimal treatment goals for each individual child and their family.

Currently, there are no standardized treatment protocols for leukemia and lymphoma for children with DNARDs. Fortunately, several similar aspects of treatment of cancer in DNARDs may be identified, while some are more specific for either AT or NBS. These aspects will be listed in the following paragraphs and can be used as starting points in choosing the right treatment strategy for each patient you may encounter. Common treatment-induced toxicities in patients with AT and NBS are summarized in Figure 2.


**We recommend that children with DNARDs and leukemia/lymphoma receive curative therapy, even in the presence of advanced cancer, if this is in accordance with patient and family wishes and the clinical condition of the child.**


In contrast to historical reports, several relatively large case series [3,4] have recently demonstrated that cure of leukemia and lymphoma in children with AT and NBS can be achieved, even with reduced-intensity therapy. The most prominent historical report has been published by Sandoval which described a cohort of 74 AT patients with lymphoid malignancies: 32 patients received chemotherapy, 50% achieved CR. Unfortunately, the follow-up was only months for most patients [99]. Dembowska published a report on 17 NBS patients in 2008 who developed NHL (9 T-LBL, 8 B-NHL). CR was achieved in 8/9 T-LBL and 3/8 B-NHL patients. All patients experienced grade IV toxicities [48]. Among 11 out of 17 patients who died, nine experienced disease progression. The median follow-up of surviving children was 3.55 (0.8–3) years.

Bienemann expanded the cohort of Seidemann and described 38 patients with DNARDs who developed hematological malignancies. 5-year OS for mature B cell lymphoma was 70%, and 48% for lymphoblastic leukemia and lymphoma. 10-year OS in the complete cohort was 58% [108]. Sandlund described five patients with mature B cell lymphoma, all treated with modified LMB-based protocol. Two achieved sustainable CR, two died of toxicity and one had progressive disease [102]. Schoenaker published in 2016 an overview of 18 AT patients with T-ALL, treated on ALL-like protocols. All but one achieved CR [14].


**Due to concerns regarding life-threatening chemotherapy-associated toxicity in patients with AT, we recommend upfront chemotherapy dose-reduction. Subsequently, dose tailoring according to individual treatment tolerance and disease response should be applied. When treating young patients with absent or very mild AT symptoms, smaller modifications or unmodified chemotherapy may be considered. For NBS patients, treatment should be initiated at ≥80% of the standard dose (SD). Delays between courses should be kept to a minimum.**


There is a decades-old controversy regarding the primary approach to chemotherapy in patients with AT. Some advocate treatment initiation according to standard treatment protocols, with subsequent dose reductions in case of toxicity. Proponents of this approach reason that there is no reliable data to indicate that patients with AT cannot tolerate standard intensive therapy, and there is also a concern that less intense therapies may not be sufficient for cure. Of the 32 patients reported by Sandoval who received chemotherapeutic treatment (between 1962–1996), 21 obtained SD and 11 received reduced dosages (RD) of chemotherapy. Median survival was short in these two groups, being 12 and 5 months, respectively. The authors concluded that those treated according to standard therapy had better chances of (prolonged) survival. However, it must be noted that 3-years overall survival was near zero [99]. The NBS pediatric patients published by Dembowska (9 T-LBL, 8 B-NHL), were all treated with upfront reduced chemotherapy. Two patients died due to treatment related toxicity. They showed however, that patients who received >80% of SD chemotherapy (*n* = 7) had better outcome (6 alive; 1 died), as compared to <80% (*n* = 10; 0 alive, 10 died). Toxicity was comparable between the groups [48].

However, since the risk of excessive side effects of chemotherapy are significant in most patients with DNARDs, several authors recommend that treatment should be initiated with reduced doses, with subsequent dose increments according to individual tolerance [14,43,102,108]. They demonstrated promising results with this approach for patients with mature B cell lymphoma as well as lymphoblastic leukemia. A total of 87 patients were treated in these studies, of whom 32 received upfront unmodified and 55 upfront modified therapy. It is noteworthy that treatment toxicity was still a significant issue even when reduced dose regimens were used [103]. In a large French cohort of AT patients treated due to cancer, 37% died of malignancies, 43% of infection, 11% of other toxicities and 80% of patients survived beyond 6 months [4].

We conclude that durable remissions can be achieved in patients with AT, even when reduced treatment doses are administered. In these studies, second malignancies were often curable as well. In contrast, data derived from the largest cohort of NBS patients who underwent chemotherapy showed that 61% died due to cancer progression, we conclude that, if possible, no, or minimal reduction of chemotherapy should be employed at the start of treatment.


**Radiotherapy leads to extreme toxicity in patients with AT or NBS and should be omitted from any treatment scheme (Level VI).**


It is clearly established that patients with DNARDs have increased radiosensitivity in vivo as well as in vitro [100,109,110,111]. Cranial radiotherapy in patients with AT leads to severe leukoencephalopathy [100] and several cases of secondary cerebral malignancies after cranial irradiation have been described [14,112]. The standard protocols for NHL/ALL do not include radiotherapy, for the minor subset of patients diagnosed with Hodgkin lymphoma (6%) we advise to omit radiotherapy.


**Careful hydration and uroprotection (mesna) are recommended during cyclophosphamide/iphosphamide therapy for all patients with AT or NBS. AT patients should receive a 50% reduction of the cyclophosphamide/iphosphamide dosage upfront (Level VI).**


Severe, late-onset and sometimes life-threatening hemorrhagic cystitis and gastrointestinal mucosal bleeding, secondary to alkylator-induced mucosal damage, have been repeatedly reported [99,113]. This is a unique toxicity for patients with AT which may be a result of mucosal damage, with impaired tissue repair and scarring with dense telangiectasias. This adverse event may be dose dependent. We recommend giving a 50% reduction upfront and escalating if treatment related complications are manageable. The standard 24 h hydration should be applied.


**Upfront reduction of high dose methotrexate (>1 g/m^2^) to 2/3 of the dose is recommended for all patients with AT or NBS who develop leukemia or lymphoma. Dose adjustment is recommended in subsequent cycles according to treatment-related toxicity. If CNS is involved, extra intrathecal MTX or three drug intrathecal regimens should be recommended.**


Since methotrexate induces the formation of DNA strand breaks due to misincorporation of uracil into DNA and inefficient DNA repair, patients with DNARDs are particularly prone to its side effects, mainly mucositis and prolonged and/or severe bone marrow aplasia [47,114]. Therefore, MTX dose reduction is the most obvious approach limiting the risk of significant toxicities. An alternative treatment strategy to reduced high-dose methotrexate in patients with ALL may be the administration of Capizzi methotrexate [104], but it is yet unclear whether this regimen will indeed result in an improved toxicity profile in children with AT.

We recommend reduction of the high dose methotrexate in ALL or mature B-NHL according to the following guidelines. At the first administration of high-dose methotrexate we recommend a dose reduction to 30% (1 g/m^2^ in case of originally 5 g/m^2^ and 2.5 g/m^2^ in case of originally 8 g/m^2^). Subsequently, the dose may be increased up to 80–100% in subsequent courses of chemotherapy, if tolerated. An alternative is the administration of Capizzi methotrexate for T-cell disease, although it is not clear whether this results in an improved toxicity profile in children with DNARDs. Furthermore, we recommend checking the level of serum folic acid prior to the administration of methotrexate, since a low level of folate relates to severe and even life-threatening skin and mucosal toxicity [115,116,117]. In order to reduce systemic toxicity of intrathecal methotrexate used to treat ALL, LBL and NHL, we recommend one dose of Leukovorin 48 h after the methotrexate injection.


**Upfront reduction of anthracycline dose is not recommended in patients with AT and NBS unless other risk factors of severe anthracycline-induced cardiomyopathy co-occur.**


Due to dosing regimens that limit total cumulative dose delivered and infrequent pre-existing cardiovascular disease, children rarely experience acute cardiotoxicity from anthracycline exposure, but they do develop chronic complications [118]. Apart from one case report describing an NBS patient who was diagnosed with fatal doxorubicin-induced cardiomyopathy [117], to the best of our knowledge, there is no data reporting the risk of anthracycline-induced toxicity among children with DNARDs. Therefore, we may assume that the contributing factors for cardiotoxicity including young age, female sex, mediastinal radiation, high cumulative dose, concomitant heart disease (including hypertension) and cyclophosphamide used in conjunction [118], are similar to the general pediatric population. Identifying co-occurrence of risk factors for anthracycline toxicity in the patient with a DNARD should involve regular monitoring with a combination of imaging and biomarkers, such as troponins and brain natriuretic peptide. In those patients with DNARDs who experience cardiac complications following anthracycline exposure or suffer from concomitant heart disease, dexarazone or liposomal doxorubicin may also be considered.

## 12. Risk of Relapse, Secondary Tumors and Overall Survival in Patients with DNARDs


**Since patients with AT or NBS have a high risk of secondary cancer each patient should be actively followed-up in a specialist pediatric oncology center (Level VI).**


Exposure to intensive chemotherapy in patients with DNARDs during the treatment of hematological malignancies was traditionally perceived to be a major factor contributing to the development of a second tumor [109]. The retrospective analysis by Klaus Bienemann showed that reduced-intensity chemotherapy did not prevent secondary malignancies [99]. Two studies performed on the largest cohort of patients with NBS published to date identified a similar finding [3,48]. Wolska et al. identified 20 (13.2%) NBS patients diagnosed with secondary malignancies, half of which occurred in patients who received an initial reduction of dose intensity [3].

## 13. The Role of Immunotherapy in Patients with Hematological Malignancies in DNARDs

Remarkable progress has been made in recent years in the field of cancer immunotherapy. Treatment options for B-cell malignancies include rituximab, blinatumomab, chimeric antigen receptor (CAR) T-cells and daratumomab for T-cell lymphoproliferation.


**The addition of rituximab to a reduced-dose chemotherapeutic backbone in the treatment of mature B-cell lymphomas is recommended.**


Current ongoing trials focus on the de-escalation of cytotoxic therapy by careful replacement of chemotherapeutic elements with targeted immunotherapy [119]. The limited published data, consisting of anecdotal case reports concerning the incorporation of rituximab into mature B-cell lymphoma therapy in children with AT and NBS [120,121,122], appears promising. Therefore, in the case of CD20 positive lymphomas, immunotherapy with rituximab might be considered, particularly when significant reductions of chemotherapy within the protocol are necessary.

The use of brentuximab vedotin in the treatment of a child with AT and Hodgkin lymphoma has been reported [123]. As neurotoxicity is well recognized following the use of brentuximab it should be used only with care in this patient group, being mindful of existing and progressive neuropathy.

Whilst no data exist in this specific patient group, the clinical efficacy of daratumumab was recently proven in relapses of T-ALL or CD19/CD22-negative ALL following allogeneic stem cell transplantation [124]. It might also be considered as a therapeutic option for patients with CD38 expressing malignancies.

Strategies preventing immune suppression and promoting immune activation, specifically immune checkpoint inhibitors, have been investigated in the management of sporadic lymphoma. This approach might be also beneficial in children with AT and NBS who develop hematological malignancies. However, there may be many immunological barriers resulting in a less effective immune response in this group of patients and careful and detailed investigations are required before these novel strategies could be widely recommended.


**Bispecific T-cell engagers (bispecific antibodies) and chimeric antigen receptor T-cell therapies (CARTs) may be considered as a bridging therapy to HSCT in NBS or as an alternative to HSCT in AT patients.**


There is a general lack of data regarding the utilization of immunotherapy in patients with immunodeficiency. Specifically, immunotherapeutic agents which rely on the patient’s immune system to be effective against cancer may not be efficacious in the context of a DNARD. Moreover, immunotherapeutic agents are still associated with notable toxicity profiles, frequently including neurotoxic events. Consequently, their applicability across a patient group with variable pre-existing neurotoxicity has yet to be established.

## 14. The Role of Hematopoietic Stem Cell Transplantation (HSCT) in AT and NBS

Primary immunodeficiencies associated with defects in non-homologous end joining (NHEJ) pathways arise because the ubiquitous repair proteins are also required to mend the physiological DNA-double strand breaks induced during T- and B-lymphocyte receptor formation as well as immunoglobulin isotype switching. Patients present with combined immunodeficiencies, bone marrow hypoplasia and susceptibility to lymphoid proliferations and frank malignancies. For many patients with combined immunodeficiencies, allogeneic HSCT enables normal immune reconstitution [125]. In order to achieve stable and durable donor stem cell engraftment, cytoreductive conditioning is employed to destroy recipient marrow cells, empty the medullary osseous niches and create space for donor stem cells. Irradiation is rarely used for patients with primary immunodeficiencies and is unnecessary. Generally, a combination of targeted busulphan with fludarabine or treosulfan with fludarabine is employed, with or without serotherapy such as anti-thymocyte globulin or alemtuzumab [126]. In conventional primary immunodeficiencies, significant ‘bystander toxicities’ may occur with patients experiencing mucositis and dermatitis as well as pulmonary, hepatic and renal toxicities. For patients with DNA-double strand breakage repair disorders, systemic toxicities are magnified by the impaired ability of cells to repair the damage caused by alkylating agents.


**Allogeneic HSCT with a modified cytoreductive chemotherapy conditioning regimen in the first remission of cancer in patients with NBS is recommended. Pre-emptive HSCT in NBS should be considered individually (Level VI).**


A large international study of patients with DNA ligase 4 deficiency, cernunnos-XLF deficiency and NBS who underwent HSCT demonstrated significantly reduced transplant-related mortality and improved survival in those patients receiving a modified cytoreductive chemotherapy conditioning regimen with reduced doses of chemotherapy agents, as compared to those receiving standard doses [127], confirming reports from smaller series of patients with NBS [13,128]. An alternative conditioning approach, for some patients at least, may be to use an antibody-based conditioning regimen to achieve myeloid engraftment [129,130].

Patients who have been successfully transplanted have demonstrated normal immunity, and resolution of bone marrow failure. Concerns remain that there may still be a residual risk of developing lympho-reticular malignancy or an increased risk of secondary malignancies, as seen in patients with Fanconi Anemia [131]. However, this has not been realized to date. In the large international series of patients transplanted for DNA ligase 4 deficiency, cernunnos-XLF deficiency and NBS there were no reported cases of lymphoma development, or the development of secondary malignancies in 212 patient follow up years [125]. A recent seminal study of 241 patients with NBS reports on lymphoma outcomes of non-transplanted and transplanted patients [3]. Forty-nine patients underwent transplant, fourteen of whom were transplanted before malignancy developed. Patients with cancer who subsequently underwent transplant had a higher 20-year overall survival than those with cancer who did not undergo transplantation and only one patient from the group transplanted preemptively developed malignancy. This study, in conjunction with that by Slack et al., suggests that there may be a beneficial effect of HSCT for patients with DNARDs, not only in restoring immunity, but on preventing the development of malignancy in this cancer-prone population.


**HSCT is not recommended in patients with AT (Level VI).**


There are few published data on outcomes of HSCT for patients with Ataxia Telangiectasia—the results are generally worse compared to those of patients with other DNA-DSB repair disorders [127]. Use of modified conditioning regimens improves the outcome [132] but it is not clear currently what role transplantation plays in the management of these patients, as the progressive neurological deterioration is likely to be unaffected by transplantation and it is not currently routinely recommended outside of clinical trials.

## 15. Screening the AT and NBS Patient’s Family


**All patients diagnosed with AT or NBS should be offered genetic counselling by a clinical geneticist preferably with experience in the field of primary immunodeficiency and cancer predisposition syndromes.**


Once two pathogenic variants are identified in an individual with clinical signs of AT or NBS, carrier testing can also be offered to unaffected family members. Both syndromes are inherited in an autosomal recessive manner and therefore parents are usually heterozygous carriers. In each pregnancy there is a 25% risk of having an affected child. Exceptionally, the pathogenic *ATM/NBN* variant might not be detected in one of the parents, due to de novo mutation, germline mosaicism or uniparental disomy in the child [133], which of course alters the recurrence risk. If both parents are carriers of a heterozygous pathogenic variant, genetic counselling regarding reproductive options is recommended. This includes natural pregnancy, prenatal testing, adoption, gamete donation and various assisted reproductive technologies such as preimplantation genetic diagnosis and in vitro fertilization (IVF-PGD). Prenatal diagnostics of AT and NBS, ideally performed by site-specific mutation analysis, is available if both pathogenic variants have been identified. Antenatal diagnosis can also be performed using haplotype analysis if an unambiguous clinical diagnosis has been made for the affected child [2]. Molecular testing can be performed on fetal DNA which is obtained either by chorionic villus sampling (CVS) or by amniocentesis. With respect to the prenatal diagnosis of NBS by ultrasound investigation, phenotypic traits including hydrocephalus and dystopic kidneys are accessory parameters which enable recognition of the syndrome [134]. However, NBS cannot be prenatally excluded even if the result of ultrasound examination is normal.

In the absence of a suggestive phenotype there is no recommendation for testing siblings of an affected proband until they reach reproductive age, but it should be considered at the parent’s request. Each sibling of the proband’s parents is at a 50% risk of being a carrier of an *ATM/NBN* pathogenic variant. If the sibling of the proband has a phenotype highly suggestive for NBS or AT, mutation analysis should be discussed [1]. Heterozygous carriers of *ATM* pathogenic variants are in general asymptomatic, but they have a reduced life expectancy because of mortality from cancer (mainly breast and digestive tract malignancy) and ischemic heart disease [64].

Because of an almost three times increased risk of developing breast cancer, heterozygous women harboring an *ATM* mutation should be offered an intensified surveillance program for breast cancer [135]. Similarly, an increased susceptibility to develop several types of malignancies (including breast and prostate cancers) is also observed in heterozygous carriers of an *NBN* pathogenic variant, especially in those harboring the c.657_661del5 founder mutation [136,137]. For heterozygous carriers of pathogenic variants in NBN, no consensus tumor screening protocols have yet been published.

## 16. Conclusions

In conclusion, cure of children with AT or NBS and leukemia and lymphoma is possible. Interpatient heterogeneity should be addressed by more precise individual treatment tailoring, with a risk-stratification system uniquely suited to this population. The optimal management of cancer in these children is unknown, mainly due to the scarcity of data regarding treatment, toxicity patterns and outcomes. Most of the available information is derived from case reports and small case series. A large-scale retrospective study performed among NBS patients revealed the beneficial role of HSCT in the first remission of cancer. Since randomized clinical trials are highly unlikely in this patient subgroup, due to this disorder’s rarity, further retrospective studies offer a valuable source of information for future decision-making. In addition, molecular biological studies of tissue samples of patients with DNARDs who developed hematological malignancies are necessary to identify potentially targetable cellular pathways. Only by international collaboration (e.g., LEGEND COST Action and Host Genomic Variation I-BFM Study Group) are we able to develop meaningful targeted or dose adapted therapy which will provide optimal quality of life for this extremely vulnerable patient group.

## Figures and Tables

**Figure 1 cancers-14-02000-f001:**
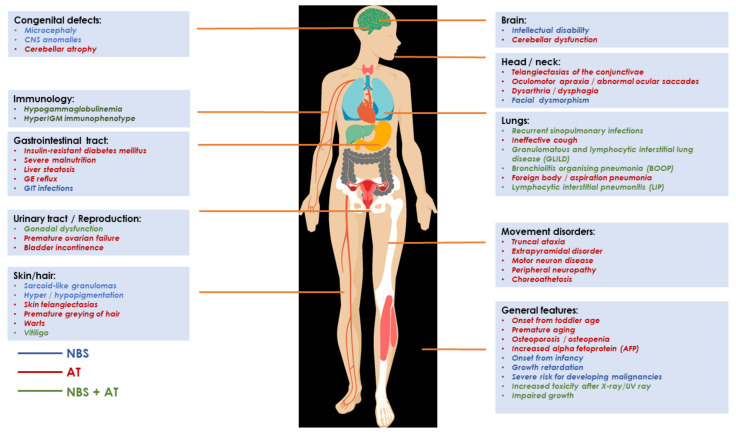
Phenotypic features of Nijmegen breakage syndrome (NBS) and Ataxia Telangiectasia (AT). The colour of the font represents specific syndrome: blue—NBS; red—AT; green—both NBS and AT.

**Figure 2 cancers-14-02000-f002:**
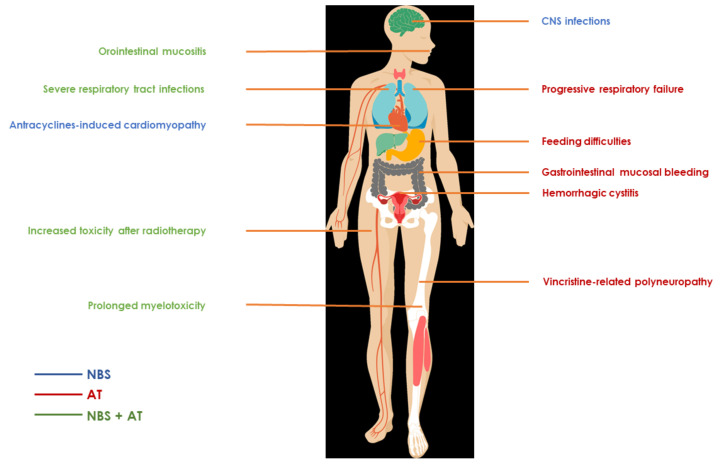
Common treatment-induced toxicities in patients with DNA repair disorders. The color of the font represents specific syndrome: blue—NBS; red—AT; green—both NBS and AT.

**Table 1 cancers-14-02000-t001:** Immunological abnormalities in patients with double stranded DNA repair disorders.

Laboratory Parameter	AT	NBS
IgG level	N/↓	N/↓
IgG2/or IgG4/or IgG1 level	↓	↓
IgM level	N/↓/↑	N/↓/↑
IgA level	Absent/↓	Absent/↓
IgE level	Absent/↓	Absent/↓
Number of CD3+ T cells	N/↓	N/↓
Number of CD4+ T cells	N/↓	N/↓
Number of CD8+ T cells	N/↓	N/↓
Number of CD19+ B cells	N/↓	N/↓
Number of CD16/56+ NK cells	N	N
Number of CD4 + CD45 RA+ cells	↓	↓
TREC and KREC levels	↓	↓
Mitogen induced T and B cell responses	N/↓	↓
Response to protein Ag	↓	↓
Response to polysaccharide Ag	↓	↓
TCR γδ T cells		

Abbreviations: TREC, T-cell recombination excision circles; KREC, kappa-deleting element recombination circle; TCR, T-cell receptor; Ag, antigen; N, normal; AT, Ataxia Telangiectasia; NBS, Nijmegen breakage syndrome.

## Data Availability

Not applicable.

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
