# Peer review of "Consensus Recommendations for the Clinical Management of Hematological Malignancies in Patients with DNA Double Stranded Break Disorders"

_cancers, 2022, doi:10.3390/cancers14082000_

Round 1

Reviewer 1 Report

In this manuscript, Pastorzczak et al performed a systematic review of patients with double stranded DNA repairs disorders (DNARD), including Ataxia Telangiectasia (AT) and Nijmegen Breakage syndrome (NBS). They discuss these disorders from different perspectives, from the diagnosis procedure, follow-up, develop of hematological malignancies and treatment options. They address these items reviewing the data available and based on that propose a recommendations.

Major revision:

- In the first part of the manuscript the authors introduced the increase risk to develop malignancies, especially hematological disorders. T-PLL is characterized by inv(14) or t(X;14) juxtaposing TCRA/D to TCL1 or TCRA/D to MTCP1, respectively. High proportion of T-PLL patients harbored ATM mutations, explaining why AT patients could evolve frequently this type of leukemia. The authors should explain in more detail this association and also explain why NBM could also develop T-PLL, which from the biological perspective is less clear.

- The authors described that different genomic variants show distinct phenotype and clinical behavior. Loss of function deletion, c. 657_661del5 are associated with classical NBS presentation, while rare missense mutations c.643C>T contributes to the severe phenotype of NBS. A modeling studies, ie. domains altered, interaction with other proteins, 3D modeling, of the different mutations would help to understand the different clinical behavior. The authors could review if there are some modeling studies performed to support these data.

- In the description of ATM mutations the authors could include additional data on functional analysis supporting the role that some atypical alterations in ATM with residual AT kinasa function lead to lees aggressive disease.

- The authors described in detail the immunological characteristics of DNARDs patients. The low proportion of IgA, IgG, and IgE could be attributed to the failure to switch from the IGM isotype to the other immunoglobulins in the germinal center. The authors could state this more in detail in the text to correlate the molecular defects with the clinical manifestations.

- The heterozygous women harboring an ATM germline mutation have three times increase risk of developing breast cancer. An obvious question not address in the review is why the patients with homozygous ATM germline mutations have higher incidence to develop hematological tumors and the heterozygous ATM mutation breast cancer.

Minor revisions:

- The authors should review that in some parts of the manuscript and figures AT is written as A-T, please unify the format.

- Section 12, there are a highlighted paragraph describing the use of CARTs and HSCT please review that this part should be here or instead in section 13.

Author Response

Reviewer #1 (Comments to the Author):

In this manuscript, Pastorzczak et al performed a systematic review of patients with double stranded DNA repairs disorders (DNARD), including Ataxia Telangiectasia (AT) and Nijmegen Breakage syndrome (NBS). They discuss these disorders from different perspectives, from the diagnosis procedure, follow-up, develop of hematological malignancies and treatment options. They address these items reviewing the data available and based on that propose a recommendations.

Major revision:

1- In the first part of the manuscript the authors introduced the increase risk to develop malignancies, especially hematological disorders. T-PLL is characterized by inv(14) or t(X;14) juxtaposing TCRA/D to TCL1 or TCRA/D to MTCP1, respectively. A high proportion of T-PLL patients harbor ATM mutations, explaining why AT patients develop this type of leukemia relatively frequently. The authors should explain in more detail this association and also explain why NBM could also develop T-PLL, which from the biological perspective is less clear.

Response: We thank the reviewer for this comment. The possible explanation is that the lymphocytes from NBS and AT patients show spontaneous rearrangements involving mainly chromosomes 7 and 14  which may lead to TCRA/D translocation. This may eventually result in the dysregulation of  TCRA/D  and clonal evolution to T-PLL. We have partly provided this information the section 3:

‘’AT and NBS patients may present with non-malignant, clonal T-cell proliferations. These proliferations recurrently harbor chromosomal rearrangements, mainly between chromosomes 7 and 14, and may evolve to aggressive T-cell prolymphocytic leukemia (T-PLL)’’.

Now, it is additionally clarified in the sentence:

‘’Equally, AT and NBS patients may present with non-malignant, clonal T-cell proliferations which harbour chromosomal rearrangements, mainly between chromosomes 7 and 14. Due to TCR dysregulation these may evolve to aggressive T-cell prolymphocytic leukemia (T-PLL).

2- The authors described that different genomic variants show distinct phenotype and clinical behavior. Loss of function deletion, c. 657_661del5 is associated with classical NBS presentation, while rare missense mutations c.643C>T contribute to the severe phenotype of NBS. Modeling studies, ie. domains altered, interaction with other proteins, 3D modeling, of the different mutations would help to understand the different clinical behavior. The authors could review if there are some modeling studies performed to support these data.

Response: We are grateful for this interesting and important suggestion. We found one study performed by Mendez G at al., in which authors compared the impact of two different pathogenic variants within NBN gene on DNA damage response. They showed  that the cleavage of the BRCT tandem domains of NBN by the 657del5 mutation affects the DNA damage response less than the Arg215Trp mutation (PMID: 22941933). This may possibly contribute to the severe phenotype which is observed in carriers of Arg215Trp mutation of NBN gene. Although the observation is valuable, we did not find any further study which would describe functional differences of these NBN variants and their correlation with phenotype in mice models. Therefore, regarding the main clinical scope of the review we decided not to discuss this study in the manuscript.

3- In the description of ATM mutations the authors could include additional data on functional analysis supporting the role that some atypical alterations in ATM with residual AT kinase function lead to less aggressive disease.

Response: We thank the reviewer for the comment. Although in this review we cannot analyzed all ATM genotype and phenotype correlations in a very detailed fashion, in paragraph 4 we now provide the reader information about the appropriate review that extensively describes this topic:

‘’There are two main categories of ATM mutations: ‘’classic’’ and “atypical” (“variant” or “mild”) [2,33].The first type includes truncating mutations leading to the loss of ATM expression and loss of ATM kinase activity. These defects are associated with severe (classic) phenotype of AT. Atypical lesions include missense mutations, in-frame or leaky splice-site mutations which generate residual amounts of functioning ATM protein with a certain level of kinase activity. In these cases, clinical presentation of AT is usually mild, and the diagnosis of cancer can precede the diagnosis of AT [2]. The current knowledge about correlations between ATM genotype and phenotype was comprehensively summarized by Putti S et al. (PMID: 34771661).’’

4- The authors described in detail the immunological characteristics of DNARDs patients. The low proportion of IgA, IgG, and IgE could be attributed to the failure to switch from the IGM isotype to the other immunoglobulins in the germinal center. The authors could state this more in detail in the text to correlate the molecular defects with the clinical manifestations.

Response: We appreciate this important comment of the reviewer and we addressed this relation more clearly in the section 6 of the manuscript.

‘’The immunodeficiency results from the impaired double-strand break repair processes that affect V(D)J recombination and immunoglobulin class-switch leading to clonally restricted immunoglobulin and T cell receptors rearrangements [56]. This eventually leads to reduced or absent serum IgA and IgG2.’’

5- The heterozygous women harboring an ATM germline mutation have a three times increased risk of developing breast cancer. An obvious question not addressed in the review is why the patients with homozygous ATM germline mutations have higher incidence to develop hematological tumors and the heterozygous ATM mutation breast cancer.

Response: We did not address the question why heterozygous ATM mutation carriers develop breast cancer because the focus of this review was homozygous (classic) patients who predominantly are diagnosed with leukemia and lymphoma. To our knowledge, there are no reliable studies which provide explanation for this finding.

Of course, we may speculate that in heterozygous carriers the expression of ATM (even if it is decreased), ensures appropriate VDJ recombination and switch of classes protecting lymphoid cells from malignant transformation.  Additionally, pathogenic germline variants in one allele of other genes involved in double strand DNA break repair including BRCA1, BRCA2, PALB2, CHEK2 highly predispose adult woman to develop breast cancer at relatively young age indicating that dysfunction of DNA repair pathways particularly promotes carcinogenesis in breast tissue.

Minor revisions:

6- The authors should review that in some parts of the manuscript and figures AT is written as A-T, please unify the format.

Response: We now unified the format in the manuscript according to the reviewer suggestion.

7- Section 12, there are a highlighted paragraph describing the use of CARTs and HSCT please review that this part should be here or instead in section 13.

Response: In the manuscript the use of CARTs in DNARDs is described in the context of HSCT, as a bridging therapy to bone marrow transplantation or as a rescue option for those patients who cannot undergo HSCT. However, CARTs as a personalized and advanced form of immunotherapy show many similar limitations of using in immunodeficient patients as compared to other immunotherapeutic drugs. Therefore, in our opinion it should be described in the chapter which addresses the possibilities and limitations of using immunotherapy in patients with AT and NBS.

Reviewer 2 Report

The authors describe the clinical features of both ataxia telangiectasia and Nijmegen Breakage Syndrome and the genetics of both disorders. This is done well, as is the description of the pathophysiology of leukaemia and lymphoma development in AT and NBS. Both AT and NBS are immunodeficiency disorders and the defects in each is also described well. The treatment of leukaemia and lymphoma in children with AT and NBS is the largest section with multiple recommendations. Although the cancer predisposition in AT and NBS is huge, the rarity of the disorders means that the absolute numbers of tumours is relatively small and the clinical heterogeneity, particularly in A-T adds to the variation. Overall, this is a good review of this rare group of patients, undertaken by a group of authors used to treating these patients and providing some well-considered recommendations. 

Comments

Line 30 Simple summary

OK, but lines 30 & 34 a jarring ‘AT and NBS are the most common DNA repair disorders’  and then a couple of lines below ‘AT and NBS are rare genetic entities’ lines .

Line 41 Abstract OK but line 46- what are ‘structural abnormalities’ here? I think this should be ‘dysmorphisms’

Line 50 – English- Change to ‘These complications make modifications necessary to commonly used treatment protocols in almost all cases.’

Line 53 ..’on the most important domains…’ What is the meaning here of domains?

I think the recommended avoidance, at present, of the use of HSCT in A-T, in contrast to NBS, should be in the abstract.

Line 60 Introduction

OK

Line 86 Article design- important and explained well. The review will provide recommendations - evidence based or expert based - and will be stated in bold type.

Line 95 Clinical phenotype of AT and NBS-

Section good. Describes well features in common.

In this section, I couldn’t see mentioned the fact that NBS patients have similar chromosomal abnormalities to A-T (although this is mentioned in section 5) and also increased sensitivity to ionizing radiation like A-T (this also comes up in section 7 (line 362), but could do with emphasis at the outset, I feel). So NBS and AT share these features. This needs to be corrected.

Line 102 - shouldn’t this really be ‘oculomotor apraxia’

Figure 1 is good

Line 147 Genotype of AT and NBS

Section good

Line 188 Pathophysiology of leukaemia and lymphoma in AT and NBS

Section good. Second part from line 215 describes well the spectrum of haematological tumours seen in these two disorders compared with those in the general childhood population and also the likely mechanisms of lymphomagenesis.

Line 249 Immunological abnormalities

AT and NBS are both immunodeficiency disorders. Similarities and differences in the clinical features are well described. This section provides three recommendations – one related to the fact that it is not known how tumour treatment affects he immune system in AT and NBS, one on antibiotic prophylaxis during tumour treatment and one on immunoglobulin replacement.

Line 336 Pulmonary manifestation in AT and NBS.

Pulmonary problems are a feature of A-T, in particular, but also NBS and recommendations for respiratory care are made for patients who develop hematological tumours.

Line 411 Nutritional status

Maintaining a good nutritional status is difficult in normal circumstances in both AT and NBS and recommendations are made in this important area.

Line 469 Neurological aspects of cancer treatment in AT and NBS.

OK but two sections 9

Line 522 Treatment of leukaemia and lymphoma in children with AT and NBS.

This is the largest section with multiple recommendations. The first is that children receive curative therapy, if that is the family’s wishes. But this would normally be given in a reduced dosage mode. Radiotherapy should be avoided totally, and special measures should be taken in treatment with cyclophosphamide and high dose methotrexate although there is no indication for reduction in dose of anthracyclines. Good

In Fig 2 – what does the black font describe?

Line 679 Risk of relapse or occurrence of second tumour

Line 692 Immunotherapy in AT and NBS

Consideration of immunotherapy is suggested particularly rituximab in CD20 positive lymphomas, and anti-CD38 monoclonal ab for some tumours. The authors are hopeful about the prospects for immunotherapy in treating tumours in these in these immunodeficiencies.

Line 732 Hematopoietic stem cell transplantation (HSCT) 

Ionizing radiation is not used for conditioning but rather some combination with fludarabine. Interestingly there is a major difference in the recommendations for AT and NBS. Allogeneic HSCT is recommended for NBS, but HSCT is not recommended in AT. I think that this difference should be mentioned in the abstract.

Author Response

Reviewer #2 (Comments to the Author):

The authors describe the clinical features of both ataxia telangiectasia and Nijmegen Breakage Syndrome and the genetics of both disorders. This is done well, as is the description of the pathophysiology of leukaemia and lymphoma development in AT and NBS. Both AT and NBS are immunodeficiency disorders and the defects in each is also described well. The treatment of leukaemia and lymphoma in children with AT and NBS is the largest section with multiple recommendations. Although the cancer predisposition in AT and NBS is huge, the rarity of the disorders means that the absolute numbers of tumours is relatively small and the clinical heterogeneity, particularly in A-T adds to the variation. Overall, this is a good review of this rare group of patients, undertaken by a group of authors used to treating these patients and providing some well-considered recommendations. 

Comments of the reviewer

1-Line 30 Simple summary

OK, but lines 30 & 34 a jarring ‘AT and NBS are the most common DNA repair disorders’  and then a couple of lines below ‘AT and NBS are rare genetic entities’ lines .

Response: Both these sentences are true : even if AT and NBS are indeed the most common DNA repair disorders, they are still rare diseases. Writing such a sentence, we aimed to provide the reason why we wrote down the review which can possibly support oncologists in treating a patient with one of these infrequent disorders.

2- Line 41 Abstract OK but line 46- what are ‘structural abnormalities’ here? I think this should be ‘dysmorphisms’

Response: We agree with the reviewer that this was unclear, we now use term ‘’congenital defects’’ instead of ‘’structural abnormalities’’.

3- Line 50 – English- Change to ‘These complications make modifications necessary to commonly used treatment protocols in almost all cases.’

Response: We corrected the sentence accordingly.

4- Line 53 ..’on the most important domains…’ What is the meaning here of domains?

Response: We agree with the reviewer that the sentence can be misleading and we removed the expression ’’on the most important domains’’ from this sentence.

5- I think the recommended avoidance, at present, of the use of HSCT in A-T, in contrast to NBS, should be in the abstract.

Response: We agree with the reviewer that the difference of the beneficial role of HSCT between AT and NBS is clinically particularly relevant. However, due to the limit of words which is required for the abstract it would be very difficult to address this specific issue there. However, because of the clinical significance, we discussed the transplantation in DNARDs in the separate chapter.

Line 60 Introduction

OK

Line 86 Article design- important and explained well. The review will provide recommendations - evidence based or expert based - and will be stated in bold type.

6- Line 95 Clinical phenotype of AT and NBS-

Section good. Describes well features in common.

In this section, I couldn’t see mentioned the fact that NBS patients have similar chromosomal abnormalities to A-T (although this is mentioned in section 5) and also increased sensitivity to ionizing radiation like A-T (this also comes up in section 7 (line 362), but could do with emphasis at the outset, I feel). So NBS and AT share these features. This needs to be corrected.

Response: We thank the reviewer for this comment, now we include the information about an increased radiosensitivity when describing common clinical features of both syndromes:

‘’Whilst clear differences exist in clinical phenotype and progression, AT and NBS also share central clinical features, notably (progressive) immunodeficiency, strong radiosensitivity and a significantly increased risk of malignancy’’.

Line 102 - shouldn’t this really be ‘oculomotor apraxia’

Response: We thank the reviewer for this suggestion, now oculomotor apraxia is mentioned as a neurological features of AT phenotype.

‘’ Additional neurological signs are oculomotor apraxia, dysarthria, postural changes, and joint contractures.’’

Figure 1 is good

Line 147 Genotype of AT and NBS

Section good

Line 188 Pathophysiology of leukaemia and lymphoma in AT and NBS

Section good. Second part from line 215 describes well the spectrum of haematological tumours seen in these two disorders compared with those in the general childhood population and also the likely mechanisms of lymphomagenesis.

Line 249 Immunological abnormalities

AT and NBS are both immunodeficiency disorders. Similarities and differences in the clinical features are well described. This section provides three recommendations – one related to the fact that it is not known how tumour treatment affects he immune system in AT and NBS, one on antibiotic prophylaxis during tumour treatment and one on immunoglobulin replacement.

Line 336 Pulmonary manifestation in AT and NBS.

Pulmonary problems are a feature of A-T, in particular, but also NBS and recommendations for respiratory care are made for patients who develop hematological tumours.

Line 411 Nutritional status

Maintaining a good nutritional status is difficult in normal circumstances in both AT and NBS and recommendations are made in this important area.

Line 469 Neurological aspects of cancer treatment in AT and NBS.

OK but two sections 9

Response: We corrected this mistake in the revised version of the manuscript.

Line 522 Treatment of leukaemia and lymphoma in children with AT and NBS.

This is the largest section with multiple recommendations. The first is that children receive curative therapy, if that is the family’s wishes. But this would normally be given in a reduced dosage mode. Radiotherapy should be avoided totally, and special measures should be taken in treatment with cyclophosphamide and high dose methotrexate although there is no indication for reduction in dose of anthracyclines. Good

In Fig 2 – what does the black font describe?

Response: We thank the reviewer for this comment. This is our mistake, now the black font is changed and refers to the legend.

Line 679 Risk of relapse or occurrence of second tumour

Line 692 Immunotherapy in AT and NBS

Consideration of immunotherapy is suggested particularly rituximab in CD20 positive lymphomas, and anti-CD38 monoclonal ab for some tumours. The authors are hopeful about the prospects for immunotherapy in treating tumours in these in these immunodeficiencies.

Line 732 Hematopoietic stem cell transplantation (HSCT) 

Ionizing radiation is not used for conditioning but rather some combination with fludarabine. Interestingly there is a major difference in the recommendations for AT and NBS. Allogeneic HSCT is recommended for NBS, but HSCT is not recommended in AT. I think that this difference should be mentioned in the abstract.

Response: We agree with the reviewer that the difference of the beneficial role of HSCT between AT and NBS is particularly relevant from the clinical point of view. However, due to the limit of words which is required for the abstract, it would be very difficult to address this specific issue there.

Reviewer 3 Report

Drs Pastorczak and colleagues provide guidelines on the clinical management of patients with AT and NBS syndrome who develop hematologic malignancies. The article design is particularly excellent and clear with recommendations in bold text followed by discussion of supporting evidence.  The multidisciplinary scope of the guidelines is appropriate as well. This manuscript will be very helpful to clinicians who do not see many patients with AT or NBS.

The authors may consider adding/ grading the level of evidence of each recommendation, either based on "Evidence Based Medicine" Levels of Evidence (example: https://www.cebm.ox.ac.uk/resources/levels-of-evidence/oxford-centre-for-evidence-based-medicine-levels-of-evidence-march-2009) or similar to the NCCN guidelines levels of evidence (https://www.nccn.org/guidelines/guidelines-process/development-and-update-of-guidelines). It does appear that most are Level 2, but this could be a way to further clarify which recommendations are based on expert opinion alone (eg: the daratumumab and bispecific and CART recommendations, page 17).

Author Response

Reviewer #3 (Comments to the Author):

Drs Pastorczak and colleagues provide guidelines on the clinical management of patients with AT and NBS syndrome who develop hematologic malignancies. The article design is particularly excellent and clear with recommendations in bold text followed by discussion of supporting evidence.  The multidisciplinary scope of the guidelines is appropriate as well. This manuscript will be very helpful to clinicians who do not see many patients with AT or NBS.

The authors may consider adding/ grading the level of evidence of each recommendation, either based on "Evidence Based Medicine" Levels of Evidence (example: https://www.cebm.ox.ac.uk/resources/levels-of-evidence/oxford-centre-for-evidence-based-medicine-levels-of-evidence-march-2009) or similar to the NCCN guidelines levels of evidence (https://www.nccn.org/guidelines/guidelines-process/development-and-update-of-guidelines). It does appear that most are Level 2, but this could be a way to further clarify which recommendations are based on expert opinion alone (eg: the daratumumab and bispecific and CART recommendations, page 17).

Response: We appreciate this suggestion of the reviewer. Due to the lack of large controlled studies performed among patients with these rare disorders, the majority of recommendations are experts opinions which based on best available literature (level VII)  and only a few of them derived from larger descriptive studies (level VI). Now we describe this in the methodology section and when the evidence level is other than VII, we indicate the level next to the recommendation.

‘’Each section defines the topic that will be addressed, followed by evidence-based or expert-based recommendations (highlighted in bold). If the level of evidence is lower than level VII (expert opinion) it is subsequently listed below the recommendation’’.